# Pharmacokinetics of Venetoclax Co-Administered with Posaconazole in Patients with Acute Myeloid Leukemia

**DOI:** 10.3390/pharmaceutics15061680

**Published:** 2023-06-08

**Authors:** Simona De Gregori, Eleonora Gelli, Mara Capone, Giulia Gambini, Elisa Roncoroni, Marianna Rossi, Claudia Patricia Tobar Cabrera, Gianluca Martini, Ludovica Calabretta, Luca Arcaini, Riccardo Albertini, Patrizia Zappasodi

**Affiliations:** 1Clinical and Experimental Pharmacokinetics Unit, Department of Diagnostic Medicine, Fondazione IRCCS Policlinico San Matteo, 27100 Pavia, Italy; m.capone@smatteo.pv.it (M.C.); r.albertini@smatteo.pv.it (R.A.); 2Department of Molecular Medicine, University of Pavia, 27100 Pavia, Italy; eleonora.gelli01@universitadipavia.it (E.G.); claudiapatrici.tobarcabrera01@universitadipavia.it (C.P.T.C.); gianluca.martini01@universitadipavia.it (G.M.); ludovica.calabretta01@universitadipavia.it (L.C.); arcaini@unipv.it (L.A.); 3Unit of Clinical Epidemiology and Biometry, Fondazione IRCCS Policlinico San Matteo, 27100 Pavia, Italy; g.gambini@smatteo.pv.it; 4Division of Hematology, Fondazione IRCCS Policlinico San Matteo, 27100 Pavia, Italy; e.roncoroni@smatteo.pv.it (E.R.); ma.rossi@smatteo.pv.it (M.R.)

**Keywords:** acute myeloid leukemia, elderly patients, venetoclax, posaconazole, HPLC-MS/MS, dose adjustment, pharmacokinetics

## Abstract

The Food and Drug Administration currently approves the combination of hypomethylating agents (HMA), azacytidine or decitabine with venetoclax (VEN) for acute myeloid leukemia (AML) patients aged more than 75 years and for patients unsuitable for intensive chemotherapy. The risk of fungal infection in the early phase of treatment is not negligible; therefore, posaconazole (PCZ) is commonly administered as primary prophylaxis. A drug–drug interaction between VEN and PCZ is well known, but the trend of serum levels of venetoclax when both drugs are overlapped is not clear. In total, 165 plasma samples from 11 elderly AML patients receiving combined treatment with HMA, VEN and PCZ were analyzed by a validated analytical method (high-pressure liquid chromatography–tandem mass spectrometry). Venetoclax trough plasma concentrations were detected during the 3 days of ramp-up as well as on day 7 and day 12 of treatment when the exposure as the area under the plasma concentration–time curve and the accumulation ratio were also calculated. The results were compared with the expected data for 400 mg/dose VEN administered alone—the confirmed high inter-individual variability in pharmacokinetics suggests the need for therapeutic drug monitoring.

## 1. Introduction

Standard treatment for most acute myeloid leukemia (AML) patients is high-intensity chemotherapy often followed by hematopoietic stem cell transplantation. However, many AML patients are ineligible to receive intensive chemotherapy (IC) because of their advanced age or comorbidities. Moreover, biology of the disease is often aggressive and chemoresistant; consequently, elderly patients are at high risk of a poor outcome, with only 2.4% remaining alive 10 years after the diagnosis [1]. The treatment for these patients has been historically challenging, sometimes with options limited to low-dose cytarabine (LDAC) or to hypomethylating (HMA) agents such as azacitidine or decitabine, obtaining a very low remission rate and short survival [2,3,4]. These discouraging results highlight the need for more effective therapies with a low toxicity profile. This option has become feasible after the advent of biological compounds exerting an alternative anti-leukemic effect to standard chemotherapy.

B-cell lymphoma 2 (BCL-2) is an antiapoptotic protein that plays a key role in the survival and therapeutic resistance of AML cells, including the leukemia stem cell (LSC) population [5,6]. BCL-2 and its family members prevent apoptosis by binding to and sequestering pro-apoptotic proteins [7]. Venetoclax (VEN) is a potent, selective and oral inhibitor of BCL-2, which, in preclinical studies, demonstrated anti-AML and anti-LSC activity as a monotherapy as well as a synergistic effect in preclinical models of AML cells when combined with azacytidine [5,6,8,9]. In patients with relapsed and refractory (R/R) AML, VEN had demonstrated modest single-agent activity [10]. In contrast, VEN at 400 mg, in combination with either azacitidine or decitabine, demonstrated significant activity in untreated patients of phase 1b [11,12] and in the difficult setting of relapsed–refractory AML patients [13]. VEN reaches the peak plasma concentration 5–8 h after an oral administration, is extensively bound to plasma proteins (>99%), has a terminal half-life of 16 to 19 h [14] and has an apparent volume of distribution (V_dss_/F) ranging from 256 to 321 L. Its exposure has a three- to five-fold increase in the presence of food and moderate/strong CYP3A inhibitors, such as the triazoles [15]. 

In the early phases of treatment, AML patients are at high risk for febrile neutropenia and life-threatening invasive fungal infections (IFI). In intensive induction chemotherapy, antifungal prophylaxis with posaconazole (PCZ) is the standard of care [16]; on the other hand, VEN-based treatments, although considered at low intensity, induce deep neutropenia, mainly at the beginning of treatment; therefore, even if it is still not clear if it is necessary, antifungal prophylaxis is also commonly adopted in less intensive regimens. In fact, in addition to the prolonged neutropenia that persists before the initiation of treatment, the hematologic toxicity due to the treatment deepens it, ultimately favoring the development of IFI. This observation led to the adoption of primary antifungal prophylaxis at least in the first cycles, before achieving remission, despite the drug–drug interactions on cytochrome P450 (CYP) enzymes and transporter proteins [17]. It is established that when VEN is co-administered with PCZ, its dose must be reduced by at least 75% [18] to ensure a safe, effective and well-tolerated dosing regimen. Most pharmacokinetic studies analyzed VEN plasma levels at different daily doses, while few studies investigated the interactions of VEN and PCZ in the AML setting. Agarval et al. studied the modifications of steady-state VEN plasma concentrations after introduction of oral PCZ, in patients with AML. They found a 7.1- and 8.8-fold increase in VEN dose-normalized C_max_ and AUC_0-24_ for the 100 mg or 50 mg daily dose, respectively, compared with the values observed when VEN was administered alone at the standard daily dose of 400 mg. However, to our knowledge, no reports are available on the raising of VEN levels in the standard schedule currently followed in real life, including PCZ started at the end of the VEN ramp-up phase. In fact, in order to establish both the efficacy and toxic potential of VEN concentrations when combined with PCZ, it would be useful to define the time necessary to achieve the steady state and the therapeutic levels of VEN. 

The time taken to reach the steady state is about five times the half-life of a drug, so VEN could take 3 or 4 days to reach its steady state—the co-administration of PCZ could affect it and lengthen the time. Moreover, both drugs have non-linear pharmacokinetics; therefore, it could be hard to predict their levels in the first phases of their co-administration. Their interference could delay the achievement of VEN therapeutic levels that are necessary to be promptly allowed in patients in the active phase of leukemia. 

The aim of this real-life study was to analyze the pharmacokinetics of 11 elderly outpatients with AML during the ramp-up period (day 1–3: 100–400 mg/day) and during the following days of treatment, when a reduced dose of VEN (70 mg/day) was administered with PCZ. 

Agarwal et al. [18] studied the interaction between VEN and PCZ in acute myeloid leukemia, 21 days after the administration of VEN alone. Their patients received 20–200 mg ramp-up treatment with oral VEN from day 1 to day 5; on days 6 through 20, the VEN dose was 400 mg. PCZ was introduced only on day 21 until day 28 (300 mg combined with 50 or 100 mg of VEN). The authors concluded that posaconazole can be used for antifungal prophylaxis in AML patients receiving a 75% reduced VEN dose.

In our study, PCZ was given on day 4 of VEN therapy, when steady-state conditions had not yet been achieved. Therefore, it was essential to confirm that the proposed dose adjustments were appropriate, safe and did not provide unexpected or dangerous underexposure. Time profiles of trough concentrations of VEN (as Log C) were analyzed during the first 7 days of treatment and on day 12.

## 2. Materials and Methods

### 2.1. Patients

The study was conducted in accordance with Good Clinical Practice guidelines and the ethical principles that have their origin in the Declaration of Helsinki [19]. The protocol and informed consent form were approved by the institutional review boards and all patients provided written informed consent before any study-specific procedures were performed.

Eleven elderly and weak outpatients (6 F; 5 M) with AML at their first cycle of HMA-VEN participated in this study; patients had a median age of 68 years (39–85); 7 cases were de novo AML and 4 patients were diagnosed with secondary AML (2 post-myeloproliferative neoplasms and 2 post myelodysplasia).

### 2.2. Drug and Dosage

Treatment consisted of a 28-day cycle including azacitidine (75 mg/m^2^) subcutaneously administered for the first 7 days, or decitabine (20 mg/m^2^) intravenously administered for 5 days, and VEN at an escalating dose in the ramp-up phase (100 mg, 200 mg and 400 mg, respectively, on days 1, 2 and 3) reduced to 70 mg/day continuously from day 4; PCZ was introduced from day 4 at a loading dose of 600 mg followed by a daily dose of 300 mg continuously from day 5 until the end of the cycle. 

### 2.3. Blood Collection

Blood samples were collected by venipuncture into potassium ethylendiaminetetraacetic acid (EDTA)-containing tubes immediately before the VEN oral administration (pre-dose) from day 1 to day 6. On both days 7 and 12, blood samples were also collected 2, 4, 6 and 8 h after the oral dose to detect VEN concentration and to evaluate the drug exposure (AUC_0-8_). 

Tubes were centrifuged at 10,400 rpm for at least 10 min at room temperature; separated plasma was then dispensed into a polypropylene tube and stored at −80 °C until the bioanalysis was carried out.

### 2.4. Chemicals and Reagents

Venetoclax (96.7%; CAS Number: 1257044-40-8; Product Number: C5160) and venetoclax-d7 (98.6%; CAS Number: 1257044-40-8 unlabeled; Product Number: C8102) were purchased from ALSACHIM (Illkirch-Graffenstaden, France). Acetonitrile (Product Number: 412,392,000–2.5 L) and formic acid (Product Number: 405,792–1 L) were obtained from Carlo Erba (Milano, Italy). Methanol (Product Number: 34,860–2.5 L), acetone (Product Number: 32,201–2.5 L) and ultrapure water (Product Number: 34,877–2.5 L) were from Honeywell (Offenbach, Germany). All chemicals were analytical grade.

Drug-free human plasma used for the preparation of calibrators and control samples was obtained from the Department of Transfusion Medicine, Fondazione IRCCS Policlinico San Matteo, Pavia (Italy).

### 2.5. Venetoclax and Venetoclax-d7 Working Solution

VEN and VEN-d7 are soluble in acetonitrile (CH_3_CN) and dimethylsulfoxide (DMSO). In total, 50 mL of a CH_3_CN:DMSO = 4:1 (*vol*:*vol*) solution (S_1_) was prepared to obtain two different standard solutions of VEN (1 mg/mL) and one standard solution of VEN-d7 (1 mg/mL), starting from ALSACHIM’s powder reference standards. An aliquot of S_1_ was added to the powders directly in the vials; VEN final concentration (1 mg/mL) was then reached by successive washings of the containers with S_1_, opportunely recovered in graduated Eppendorf tube. 

Two different working solutions of VEN (100 µg/mL) were then obtained by adding 900 µL of acetonitrile to 100 µL of standard solution (dilution 1:10); the procedure was repeated twice, in two separate 1.5 mL microcentrifuge tubes.

The working solution of VEN-d7 (1 µg/mL) was obtained by diluting 40 µL of its standard solutions (1 mg/mL) with a 40 mL solution of CH_3_OH: CH_3_CN = 50:50 (*vol*:*vol*), acidified with HCOOH (final concentration: 0.1%).

All the working solutions were prepared fresh daily.

### 2.6. Calibration Standards and Quality Control Preparation

Eight calibrators (U, A, B, C, D, E, F and G) were prepared in drug-free human plasma by serial dilutions: the most concentrated calibrator U (5000 ng/mL) was obtained by adding 100 µL of 100 µg/mL VEN to 1900 µL plasma in a 5 mL Eppendorf tube. The other calibrators (from 2500 to 39.06 ng/mL) were prepared by mixing 300 µL of plasma with 300 µL of the previous sample, in 1.5 mL Eppendorf tubes. The second VEN working solution (100 µg/mL) was used to obtain the quality controls (QCs) at three different levels: QcH (4000 ng/mL), QcM (400 ng/mL) and QcL (40 ng/mL), by serial dilution (1:10) in plasma.

The first one (QcH) was prepared by diluting 80 µL of the second 100 µg/mL VEN working solution with 2 mL of plasma in a 5 mL Eppendorf tube.

### 2.7. Sample Preparation

Just before the analysis, patient plasma samples were thawed and calibrators and quality controls were prepared fresh in the correct matrix. Each sample was processed by protein precipitation; 50 µL of plasma was prepared in pre-labelled Eppendorf tubes. Subsequently, 200 µL of internal standard working solution (IS: venetoclax-d7, 1 µg/mL in CH_3_OH:CH_3_CN = 50:50, 0.1% HCOOH) was added to promote sample cleanup, followed by vortexing (1 min) and centrifugation (10,400 rpm—10 min). In total, 200 µL of the supernatant was transferred to HPLC vials and 5 µL was injected into the chromatographic column. 

### 2.8. HPLC-MS/MS Assay

The HPLC system used was a Thermo Scientific quaternary pump (Accela pump, Thermo Fisher Scientific, San Francisco, CA, USA) interfaced with an autosampler (Accela autosampler). The analytical column was a Zorbax SB (C18, 4.6 × 75 mm; 3.5 µm), heated at 40 °C. The mobile phases consisted of acidified H_2_O (mobile phase A) (0.1% HCOOH) and a solvent mixture of CH_3_CN and CH_3_OH (50:50 *vol*:*vol*; 0.1% HCOOH: mobile phase B) eluted in gradient mode; the flow rate was 600 µL/min. From 0 to 1.5 min, mobile phases were 70:30 (A:B) at 1.60 min until 6.0 min 100% B. From 6.10 to 7.00 min, a mix of CH_3_CN:CH_3_OH:acetone (45:45:10 = *vol*:*vol*:*vol*) was eluted through the chromatographic column. From 7.10 to 9.00 min, the column was again equilibrated to the initial conditions. Although both VEN and IS eluted at 3.4 min, the analytical runtime was 9 min. Mass analysis was performed using a TSQ Quantum Access mass spectrometer system (Thermo Scientific, San Francisco, CA, USA) equipped with an electrospray interface and operated in the positive ionization mode, following the transition *m*/*z* 868.1 >> 320.7; 635.5 and *m*/*z* 875.2 >> 320.8; 642.9 for VEN and IS, respectively. A total runtime of 9 min was required for an adequate washing procedure of the chromatographic column; with a shorter runtime, we observed the carryover of unwanted late-eluting molecules in patients’ plasma samples. 

### 2.9. Analytical Assay

All the main bioanalytical method characteristics essential to ensure the acceptability of the performance and the reliability of analytical results (i.e., selectivity, carryover, lower limit of quantification LLOQ, calibration range, accuracy and precision and matrix effect) were analyzed over 5 days. 

After defining all of the chromatographic parameters and mass spectrometer settings, selectivity (specificity) was evaluated by separately analyzing the mobile phases (A and B), the precipitant solution containing IS and six different drug-free extracted plasma lots from healthy volunteers. The absence of carryover was also assessed by injecting mobile phase B after the calibrator “U”.

Accuracy and precision were evaluated by analyzing 5 replicates at the lower limit of quantification and at the three quality control levels (Qc): low, medium and high. Within-run accuracy (%) and precision (CV%) were calculated in a single run, whereas both the between-run accuracy and precision were calculated in different runs, on different days.

The accuracy at each concentration level had to be within ±15% of the nominal concentration, except at the LLOQ, where it could be within ±20%. The precision (CV%) could not exceed 15%, except at the LLOQ, where it should not exceed 20% [20].

The matrix effect was quantified using six different lots of drug-free human plasma from individual donors. The matrix factor was investigated at low and high concentration levels (QcL and QcH) in six drug-free human plasmas and in two specific matrices: hemolyzed and lipemic plasma samples. The matrix factor (MF) was calculated as ratio between the VEN peak area in presence of matrix (by analyzing blank matrix spiked with the working solutions) and the peak area in absence of matrix (water instead of plasma). The IS-normalized MF was also calculated by dividing the MF of the analyte by the MF of IS. The overall CV% obtained for the concentration should not be greater than 15%.

Each analytical run consisted of a zero sample (θ_IS_ blank plasma sample spiked with IS), eight calibration standards (U-G) and three quality controls (high, medium and low). Every single day, patients’ plasma samples were analyzed with a just prepared analytical run. Calibration standards were acceptable if the back-calculated concentrations were within 20% of the nominal concentration at the intended lower limit of quantification (39.06 ng/mL) and within 15% of the theoretical concentration for all other levels. Quality controls were acceptable if the calculated concentrations were within 15% of the theoretical value. An analytical run was considered acceptable when two-thirds of the quality controls analyzed in the run and at least 50% of the quality control results analyzed for each concentration level were within the acceptance limits.

### 2.10. Data Analysis

Xcalibur 2.07 and LCquan 2.5.6 software from Thermo Scientific (San Jose, CA, USA) were utilized for the LC-MS/MS system control, data acquisition and data analysis. Calibration curves were generated using weighted (1/x^2^) linear regression curves. VEN was identified with a combination of retention times and specific MRM transitions; the corresponding amounts were quantitated by normalizing the peak area to the internal standard, and concentration was calculated from the respective calibration curves.

### 2.11. Pharmacokinetics Analysis

To evaluate the impact of PCZ on VEN exposure, we monitored plasma VEN concentrations immediately before the next oral dose (C_trough_) on both the 3 consecutive days of VEN ramp-up (100, 200 and 400 mg/day) and 9 days after the co-administration of PCZ (day 12), in order to evaluate the VEN accumulation profile. Accumulation was verified as ratio between C_trough_ at the steady state (C_τss_, on day 12) and C_trough_ on day 1 (C_τ1_).

We also calculated the exposure to VEN for each patient, as the area under the plasma drug concentration–time curve (AUC_0-8_) both on days 7 and 12, using the non-compartmental model, by the linear trapezoidal rule. We compared AUC_0-8_ obtained for each patient on these days and calculated the index of accumulation (accumulation ratio, AR) by dividing AUC_0-8_ on day 12 by AUC_0-8_ on day 7. Assuming that the steady state was reached on day 12 [18], AUC_0-24_ was calculated by approximating C_24h_ equal to C_trough._ Determination of C_12h_ and C_24h_ was not included in the study because of the poor health of the elderly outpatients involved in the study. 

The apparent clearance (CL/F, where F is the bioavailability factor) of VEN was also calculated by the equation CL/F = (Daily Dose)/(AUC(0-24)).

### 2.12. Statistical Analysis

Data analysis was performed with the STATA statistical package (release 13.1, 2014, Stata Corporation, College Station, TX, USA). The agreement of AUC_0-24_ between day 7 and day 12 was assessed by Lin’s Concordance Correlation Coefficient (CCC) with its 95% confidence interval (95% CI) [21].

## 3. Results

### 3.1. Chromatography

The absence of interfering components at the expected retention times was verified for the mobile phases (A and B), the precipitant solution containing IS and six drug-free human plasmas; all responses were always less than 20% of the LLOQ and 5% for IS. Carryover was absent too.

Linearity was confirmed in the concentration range 39.06–5000 ng/mL; each curve exhibited consistent linearity and reproducibility in the specific concentration range. Regression coefficients (r) were always higher than 0.99.

All results for the within- and between-run accuracy and precision of the analytical method are reported in Table 1 and met the acceptance criteria. 

The coefficients of variation (CV%) of IS-normalized MF calculated for QcL and QcH were 7.1 and 2.7%, respectively, in accordance with the required criteria.

Figure 1 reports a representative chromatogram of VEN at the lower limit of quantification (LLOQ) (39.06 ng/mL) and VEN-d7 (IS) at its working concentration (1 µg/mL). The selected transitions and the corresponding collision energies (C.E) are also specified.

### 3.2. Venetoclax Concentration Time Profile

We analyzed 165 plasma samples; the first 55 were collected during the VEN dose ramp-up (100, 200 and 400 mg/day) over 3 consecutive days, immediately before the next dose was administered (Cτ). The other 110 were analyzed on day 7 (five samples/patient) and on day 12 (five samples/patient), just before the next oral dose (Cτ) and 2, 4, 6 and 8 h later.

We observed a small continuous increase in the mean VEN C_τ_ throughout the observation period and a wide inter-individual variability, expressed as the percentage of the coefficient of variation (CV%), on each treatment day, with a minimum on day 12 (61%) and a maximum on day 4 (94%), corresponding to the maximum administered dose of VEN alone (400 mg, on day 3). 

The highest mean trough concentration (C_τ_) was reached on day 12, although the dose of VEN was reduced by 82.5% (70 mg/day) already from day 4 (Figure 2), as a consequence of the drug–drug interaction with PCZ. The accumulation ratio between day 12 and day 1 (AR: C_τ,12_/C_τ,1_) was 11.8 (81.2%).

The mean VEN plasma concentrations on day 12 were greater (median: 36%) than the corresponding ones on day 7 (Figure 2 and Figure 3). 

Two hours after the oral administration, there was a modest plasma concentration decrease (C_2_) both on day 7 and day 12, followed by a continuous slight increase; the maximum concentration was achieved at 8 and 6 h, respectively. 

The maximum coefficient of variation (CV%) was observed at C_0_ (63%) and at C_6_ (65%) on day 7 and day 12, respectively.

VEN AUC_0-8_ increased for each patient from day 7 to day 12, as reported in Figure 4.

The increase was less than 15% for 45% of the patients (ID 3, 4, 5, 6 and 11), approximately 28% for 1 patient (ID 7), approximately 50% for 2 patients (ID 1 and 2) and about 75% for 107 and 203% for the last 3 patients (ID 10, 8 and 9, respectively). 

The median increase was approximately 28%.

The accumulation ratio (AR, AUC_0-8_) was greater than 1 for all patients except patients 6 and 11 (Table 2) due to an expected increased VEN exposure on day 12.

On the basis of these results, we observed that there was VEN accumulation for 9/11 patients from day 7 to day 12. 

Agarwal et al. reported their steady-state results after administration of 400 mg VEN alone as a mean dose-normalized AUC_0-24_ [18]. We compared our results with theirs (Table 3). 

For our study, the 24 h post-dose concentration values were not available on day 12; therefore, the pre-dose concentrations (C_τ_) on the same day were assumed to be equal to C_24h_. 

The mean calculated C_ss_ and AUC_(0-24)_ were 2046 ng/mL and 49,092 ngxh/mL/mg (CV%: 54), respectively; the expected AUC_(0-24)_ [18] was about 38,800. 

The mean apparent clearance (CL/F) of VEN, co-administered with PCZ, was 1.4 L/h (CV%: 54).

## 4. Discussion

A combination of azacitidine or decitabine and venetoclax, at a daily dose of 400 mg after a ramp-up phase, is approved for the treatment of acute myeloid leukemia (AML), even for elderly and fragile patients. Patients with AML and other hematologic malignant tumors are at high risk for infectious complications, such as invasive fungal infections (IFI). Azole compounds (e.g., posaconazole: PCZ) are commonly used as antifungal prophylaxis, especially in the initial phase of treatment, when deep neutropenia is present and the risk of infection is high. However, PCZ is responsible for a well-known drug–drug interaction because it is a strong CYP3A4 inhibitor and contributes to increase the VEN plasma concentration.

Due to the risk of tumor lysis syndrome, a dosing regimen featuring 3-day ramping of VEN alone is recommended, followed by the introduction of PCZ therapy on day 4. At the same time, the daily VEN dose must be reduced by at least 75% in order to minimize eventual side effects due to too high plasma concentrations of VEN. However, the kinetics of the raising VEN levels when PCZ is added are not known, and the time necessary to achieve therapeutic levels of VEN is not yet well defined. It is also crucial that, during an induction regimen, both therapeutic and toxic levels of drugs are predictable. Our study investigated VEN plasma levels during the co-administration of VEN and PCZ within the first cycle of HMA-VEN for AML patients.

During the first 3 days of dose escalation of VEN alone, as well as from days 4 to 12 in the presence of PCZ, we observed a small continuous increase in the mean C_τ_ of VEN. The data show a large inter-individual variability (CV%), with a minimum on day 12 (61%: steady state achieved) and a maximum on day 4 (94%). The highest mean C_τ_ was reached on day 12, although the dose of VEN was reduced by 82.5% (70 mg/day) because of the drug–drug interaction with PCZ. Venetoclax has a median terminal elimination half-life of 16-19 [14] hours; therefore, we supposed that VEN plasma concentrations achieve the steady state conditions in 3 or 4 days of treatment; we verified a delay in reaching the steady state due to the co-administration with PCZ. Trough samples, collected before the steady state had been reached, showed VEN accumulation from day 1 to day 12, providing valuable information about the drug disposition. In our study, VEN concentrations on day 7 were about 74% of the corresponding one on day 12, at the steady state; analogously, AUC_0-8_ increased by 34% from day 7 to day 12 (median AUC_0-8_: 10,649 vs. 14,685 ngxh/mL).

Comparing the AUC_0-8_ between day 12 and day 7, the calculated median accumulation ratio was 1.3 (range: 1.0–2.9). These data suggest that on day 7, the accumulation process was still ongoing, and VEN concentrations did not yet achieve the steady state. The calculated exposures give more precise information than the single measured concentrations and reduce the inter-individual variability in the results.

Agarwal et al., on a different schedule of the drug combination, observed that compared with 400 mg of VEN alone (day 20), co-administration of 50 mg of VEN with PCZ increased the mean VEN C_max_ and AUC_0-24_ by 53% and 76%, respectively. In addition, relative to 400 mg of VEN alone, co-administration of 100 mg of VEN with PCZ increased the mean VEN C_max_ and AUC_0-24_ by 93% and 155%, respectively.

In our study, a fair agreement for AUC_0-8_ was found between day 7 and day 12 (CCC = 0.65; 95% CI: 0.39–0.91). On day 12, the mean calculated AUC_0-24_ was 49,092 ngxh/mL (CV: 54%), comparable with the intermediate values between those recommended after a therapeutic dose of 400 mg once daily alone and after the co-administration of 50 mg VEN with PCZ [18], suggesting the absence of undesirable underexposure. The mean dose-normalized AUC_0-24_ (hxng/mL/mg) increased by 7.2 times, with an oral dose 5.7-fold reduction.

The effects of PCZ on CYP enzymes and transporter proteins lead to a decrease in the apparent clearance (CL/F) of VEN, which in our study was 1.4 L/h after 70 mg/day of VEN. These data are comparable with those obtained by Agarwal et al. [18] after doses of 50 and 100 mg/day in combination with PCZ: 1.1 and 1.3 L/h, respectively. Clearance was 10.3 L/h when VEN was administered alone (400 mg/day). Reduced clearance corresponds to higher plasma levels.

During the ramp-up period, we observed a slight linear increase in C_τ_ as a function of the administered doses and a substantial inter-individual pharmacokinetic variability (N = 11; CV%: 61–94), which slightly decreased over the following 9 days (CV%: 47–65). Moreover, due to interaction of the two drugs, we cannot confirm our preliminary hypothesis that the steady state could be reached after 7 days of treatment because of the drug–drug interaction with PCZ. In conclusion, plasma VEN levels measured in the early phase of AML treatment when a strong CYP3A4 as PCZ was added show a substantial inter-individual pharmacokinetic variability and a delay of steady state achievement compared to the schedule without PCZ. However, at day 12, almost all cases showed high values of drug exposure, suggesting the achievement of therapeutic levels at that time; within the first 12 days of therapy, in the presence of PCZ, TDM is an important tool in order to promptly modulate the dosages if too high and toxic plasma VEN levels are detected, even though the steady state has not yet been reached. In conclusion, proactive TDM may be a reasonable approach to improve the clinical outcome of patients treated with VEN.

Due to the limited number of studied patients and the considerable variability observed, these data need to be confirmed in a larger population.

A limitation of this study could be the assumption that on day 12, C_24_ and Cτ were thought of as reasonably similar to be used for AUC_0-24_ calculations. However, as previously reported, no patients were hospitalized; therefore, their fragile health conditions led us to this approximation—we are grateful for their participation.

## Figures and Tables

**Figure 1 pharmaceutics-15-01680-f001:**
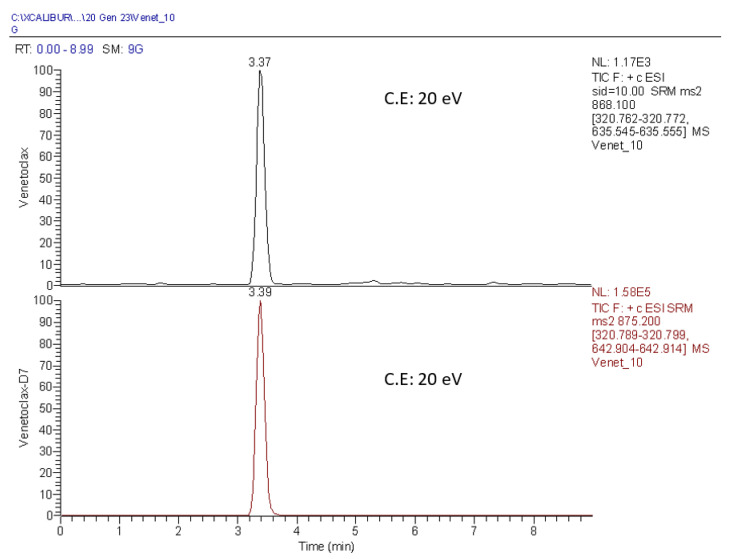
VEN and VEN-d7 chromatogram.

**Figure 2 pharmaceutics-15-01680-f002:**
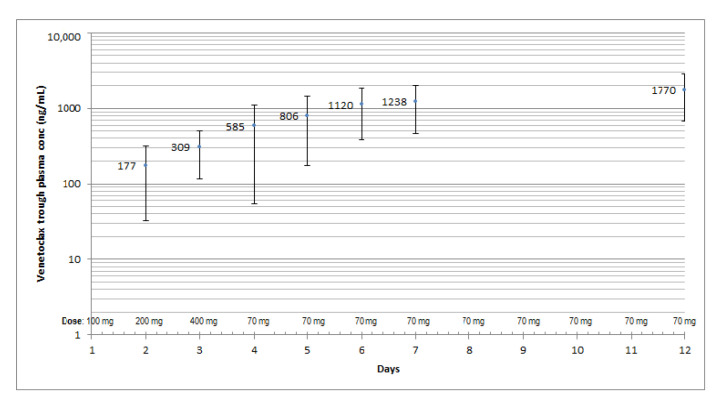
mean venetoclax (± SD) trough concentration (C_τ_) during the first 12 days of treatment.

**Figure 3 pharmaceutics-15-01680-f003:**
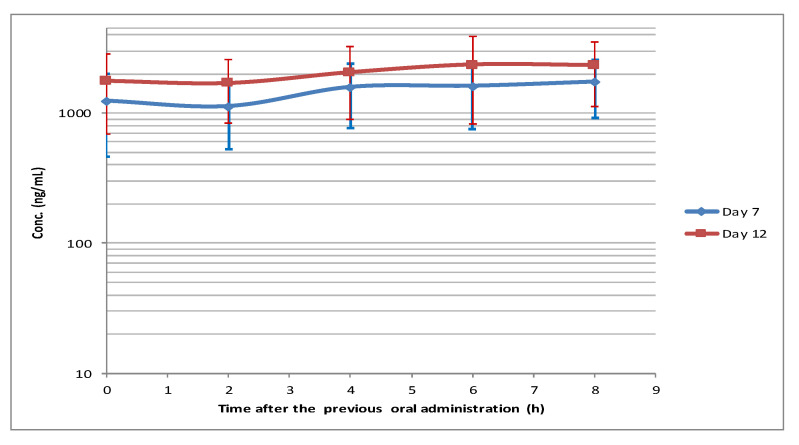
mean (±SD) venetoclax concentration profiles on day 7 and day 12.

**Figure 4 pharmaceutics-15-01680-f004:**
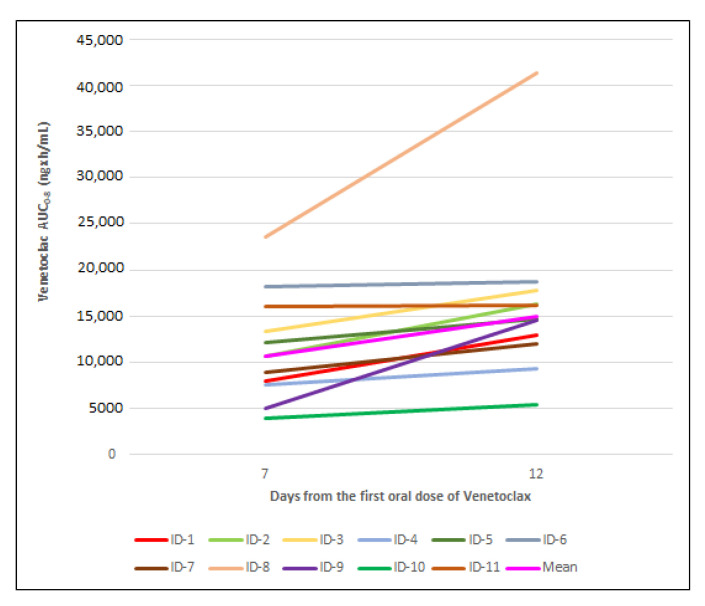
Individual venetoclax AUC_0-8_ values on day 7 and day 12.

**Table 1 pharmaceutics-15-01680-t001:** Within- and Between-run accuracy and precision.

	Nominal Concentration (ng/mL)				Linearity (ng/mL)
	LLOQ (39.06)	QcL (40)	QcM (400)	QcH (4000)	
Within-run accuracy (%)	102.8	109.6	95.4	103.7	39.06–5000
Between-run accuracy (%)	101.8	107.8	98.1	92.8	
Within-run precision (CV%)	5.1	5.5	6.5	5.3	
Between-run precision (CV%)	4.3	5.3	6.1	4.5	

LLOQ: Lower limit of quantification; QcL, QcM, QcH: Quality control levels; low, medium and high.

**Table 2 pharmaceutics-15-01680-t002:** VEN accumulation ratio (AR (AUC_0-8_)) between day 7 and day 12.

Patient ID	AUC_(0-8)_ (Day 7)	AUC_(0-8)_ (Day 12)	Accumulation Ratio
	ngxh/mL		
1	7967	12,941	1.62
2	10,650	16,265	1.53
3	13,386	17,783	1.33
4	7650	9351	1.22
5	12,144	14,685	1.21
6	18,177	18,805	1.03
7	8961	11,994	1.34
8	23,645	41,381	1.75
9	5088	14,624	2.87
10	3942	5501	1.40
11	16,073	16,246	1.01
**Mean (CV%)**	11,608 (51)	16,325 (56)	1.5 (35)
**Median**	**10,650**	**14,685**	**1.3**
**Range**	3942–23,645	5501–41,381	1.01–2.87

**Table 3 pharmaceutics-15-01680-t003:** Venetoclax Pharmacokinetic Parameters on day 12.

	ID-1	ID-2	ID-3	ID-4	ID-5	ID-6	ID-7	ID-8	ID-9	ID-10	ID-11	Mean	Expected ***	Mean/Exp.
AUC_(0-24)_ (ngxh/mL)	38,930	49,756	54,738	27,512	38,268	62,354	40,584	120,727	43,564	17,164	46,421	49,092	38,800	1.3
AUC_(0-24)norm_ (ngxh/mL)/mg *	556	711	782	393	547	891	580	1725	622	245	663	701	97	7.2
C_ss_ (ng/mL) **	1622	2073	2281	1146	1594	2598	1691	5030	1815	715	1934	2046		

* AUC_(0-24)norm_ (ngxh/mL)/mg * = AUC_(0-24)_ (ngxh/mL)/70 mg; ** C_ss_ (ng/mL) average plasma concentration at the steady state on day 12; *** Value expected after administration of VEN alone (400 mg).

## Data Availability

The original data can be requested privately from the corresponding authors.

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
