# Peer review of "Pharmacokinetics of Venetoclax Co-Administered with Posaconazole in Patients with Acute Myeloid Leukemia"

_pharmaceutics, 2023, doi:10.3390/pharmaceutics15061680_

Round 1

Reviewer 1 Report

This work is interesting with novel objective and the authors have done a number of experiments to establish their hypotheses. Still the reviewer has some queries and suggestions as follows:

1.      I propose to include the chromatographs in the manuscript

2.      In general, it is well written, but there are some grammars and typing/spacing errors should be corrected. The English is generally satisfactory but a native speaker should read the paper and correct some sentences

In general, it is well written, but there are some grammars and typing/spacing errors should be correctedThe English is generally satisfactory but a native speaker should read the paper and correct some sentences

Author Response

Dear Reviewer thank you for your suggestions.

Please consider my attachment.

Best regards

Reviewer 2 Report

The submitted manuscript is a good report on the pharmacokinetic analysis of patients with acute myeloid leukemia (AML) during the ramp-up period and treatment days when a reduced dose of venetoclax (VEN) is co-administered with posaconazole (PCZ). The authors have covered important aspects of sample collection and preparation and the necessary tests.

In the introduction section, the authors have claimed that their study is the first in their field. However, looking at the literature reveals a few similar studies. The most similar is the research done by Agarwal et al. entitled "Management of Venetoclax-Posaconazole Interaction in Acute Myeloid Leukemia Patients: Evaluation of Dose Adjustments. Clinical therapeutics" referred to the manuscript by number 18. Here the details that distinguish the study from the similarities must be highlighted.

The results are completely described. Still, comparing the results with the previous studies and describing which achievement was added will be better.

There are a few pitfalls in the text which must be corrected. See the details below.

Line 47- VEN and Line 64- PCZ

Write the full form before the first abbreviation. (Although mentioned in the abstract, it must also be written in the body).

Line 69- CYP:

Write the full form before the first abbreviation.

Several References are missing in the manuscript.

Line 81 and Line 92- and Line 98: Refer to appropriate references.

Reference number 19 seems incomplete.

Author Response

Dear Reviewer thank you for your suggestions.

Please consider my attachment.

Best Regards

Author Response

(The authors gave the same response as above.)

Round 2

Reviewer 2 Report

Accepted in the present form

Author Response

Dear Reviewer,

thank you for your support and your approval.

Reviewer 3 Report

Thank you for addressing my earlier comments. I think in general you did well on improving the paper. However, I still struggle much with the exact aim of the study and how the current study is set up to answer this question. For example: in the introduction the aim is stated as 'to analyze the pharmacokinetics'. However, in the methodology section (2.6) the primary aim is the accumulation of VEN in presence of PCZ. So if the accumulation is the primary endpoint, then why is this something one would particulary be afraid of? Also, you state that there is a knowledge gap regarding the PK of VEN in presence of PCZ in the first days of treatment (I agree), but why then analyze the accumulation at days 7 and 12? The authors state in the answers to my question that they only wanted to look at day 12 because earlier steady state would not be reached. But I do not understand this when the aim of the study is to characterize the PK during ramp up (i.e. in the first days of treatment)? In the discussion I miss an overall conclusion. What is the answer to the primary questions that were introduced in the introduction? Bottom line is, that I would encourage the authors to carefully reread the paper and try to incoorporate a clear common thread that connects the study aim with how this data is gathered and analyzed to end with a clear conclusion with regard to the identified knowledge gaps.

Lastly, the whole paper should be carefully checked on interpunction, double spaces and overall English especially where the text was thoroughly edited in the seond version. There are a lot of errors/typo's throughout the text.

I very much hope that my comments aid the authors in improving this manuscript. I still feel that there is need for more studies on the VEN-PCZ DDI and this data can be of much aid in improving what we know.

Few additional suggestions (line numbers are from the version without track changes):

- Remove lines 101-103 as it is a doublure from what is stated before.

- Line 92-99: Although I understand some parts were added based on my last comments, this part is now a little difficult to read. This part contains part of the aim of the study, but also some methodology (time profiles were analyzed...). The Agarwal study is introduced here somewhat 'out of the blue'. I would ask to rephrase this part by first introducing what is known, then what this study tries to add.

- I still think the whole thorough description of the assay and validation results of the assay is too much for the main manuscript. I would prefer to move this to a supplemental file for the few readers that are interested in this. This I would also like to leave up to the editor to decide in this with regard to the journal's policy.

See my comments and suggestions
